# Cell Architecture-Dependent Constraints: Critical Safeguards to Carcinogenesis

**DOI:** 10.3390/ijms23158622

**Published:** 2022-08-03

**Authors:** Komal Khalil, Alice Eon, Florence Janody

**Affiliations:** 1i3S—Instituto de Investigação e Inovação em Saúde, Universidade do Porto, Rua Alfredo Allen, 208, 4200-135 Porto, Portugal; kkhalil@i3s.up.pt (K.K.); alice.eon@etu.u-paris.fr (A.E.); 2IPATIMUP—Instituto de Patologia e Imunologia Molecular da Universidade do Porto, Rua Dr. Roberto Frias s/n, 4200-465 Porto, Portugal; 3Master Programme in Oncology, School of Medicine & Biomedical Sciences, University of Porto (ICBAS-UP), Rua Jorge Viterbo Ferreira 228, 4050-513 Porto, Portugal; 4Magistère Européen de Génétique, Université Paris Cité, 5 Rue Thomas Mann, 75013 Paris, France

**Keywords:** cell shape, cytoskeleton macromolecules, cell-intrinsic forces, signalling networks, cell fate, carcinogenesis, Waddington’s landscape

## Abstract

Animal cells display great diversity in their shape. These morphological characteristics result from crosstalk between the plasma membrane and the force-generating capacities of the cytoskeleton macromolecules. Changes in cell shape are not merely byproducts of cell fate determinants, they also actively drive cell fate decisions, including proliferation and differentiation. Global and local changes in cell shape alter the transcriptional program by a multitude of mechanisms, including the regulation of physical links between the plasma membrane and the nuclear envelope and the mechanical modulation of cation channels and signalling molecules. It is therefore not surprising that anomalies in cell shape contribute to several diseases, including cancer. In this review, we discuss the possibility that the constraints imposed by cell shape determine the behaviour of normal and pro-tumour cells by organizing the whole interconnected regulatory network. In turn, cell behaviour might stabilize cells into discrete shapes. However, to progress towards a fully transformed phenotype and to acquire plasticity properties, pro-tumour cells might need to escape these cell shape restrictions. Thus, robust controls of the cell shape machinery may represent a critical safeguard against carcinogenesis.

## 1. Introduction

Animal cells exhibit great diversity in their shape. Cell shape is often regulated by cell fate determinants, in order to adapt cell geometry to a particular cellular function [1,2]. However, the regulatory link between cell fate determinants and cell shape is not unidirectional. Changes in cell geometry can also affect the intrinsic signalling state of the cell, override extrinsic biochemical factors, and consequently drive cells to acquire distinct behaviours [3,4]. Cell shape is determined by the mechanical balance between intracellular and extracellular forces exerted on the cell membrane. Extrinsic forces include gravity, shear stress or tensions exerted by neighbouring cells through cadherin-mediated adherens junctions (AJs) and by the extracellular matrix (ECM), through integrin transmembrane receptors. We refer to a number of excellent reviews which highlight ground-breaking studies, providing hints into the mechanisms by which extrinsic forces affect cell shape and consequently the behaviour of normal and cancer cells [4,5,6]. In this review, we discuss the role of cell shape as a direct readout of intrinsic forces exerted by the cytoskeleton on the cell membrane in normal, pro-tumour and cancer cells. We first focus on how the organization and dynamics of the cytoskeleton macromolecules, including microfilaments (MFs), microtubules (MTs) and intermediate filaments (IFs), control cell shape by exerting forces on the plasma membrane. We then review representative examples that implicate cell shape changes in cell fate decisions and provide illustrations of how changes in cell architecture alter the transcriptional program. Finally, we discuss how cell shape could act as a critical safeguard to tumour initiation and progression and how, in turn, cancer cells could escape the restrictions imposed by cell shape to undergo malignant transformation and acquire plasticity properties.

## 2. Cell-Intrinsic Mechanisms of Cell Shape Control

In mammalian cells, the geometrical information of the space occupied by cells can be classified into three main types. The lymphoblast-like type defines spherical cells that can grow in suspension. In contrast, cells with fibroblast-like and epithelial-like shapes are anchored to ECM extracellular macromolecules, such as collagens and fibronectins, through integrins, which transmit signals via inside-out and outside-in signalling. While fibroblast-like cells are flat, spindle-shaped, epithelial-like ones are prism-shaped and are associated with each other through AJs. These major types can be further divided into subtypes. For instance, cells with epithelial-like shapes are subdivided based on their thickness. The squamous epithelial cell type has a flattened shape (surface area wider than their height), whereas cuboidal epithelial cells have similar height and width and columnar epithelial ones are taller than they are wide [7]. Yet, cell shape does not appear to be a continuous variable, as the range of shapes that a cell can acquire seems to be restricted. Systematic gene inhibition using RNA interference has shown that *Drosophila* haemocytes transit between five pre-existing shapes in a switch-like manner but never acquire stable intermediate shapes distinct from the pre-existing ones [8]. This behaviour suggests that genetic alterations attract cells into a set of defined shapes, reminiscent of the behaviour of stable attractors of signalling regulatory networks, dragged into specific phenotypes (see below basins of attraction) [9].

Cell-intrinsic forces are mostly the direct result of crosstalk between membrane-associated lipids and proteins at the plasma membrane and the force-generating capacities of the cytoskeleton [10]. The cytoskeleton is a complex intermingled meshwork of three major classes of filamentous macromolecules: MFs, MTs and IFs. These three cytoskeletal macromolecules mainly differ in their mechanical stiffness, the dynamics of their assembly, their polarity, and the type of associated molecular motors [11,12,13,14,15].

MFs are semi-flexible 7–9 nm diameter filaments composed of monomeric actin subunits arranged head to tail to give the filament a molecular polarity. Polymerization occurs predominantly by extension of the fast-growing barbed ends of the filaments, while filaments are disassembled by the loss of monomers from the slow-growing pointed ends [16]. Actin filaments generate the forces necessary to change shape through distinct mechanisms. One of those mechanisms involves the addition of actin monomers to filaments oriented towards the plasma membrane that can push the cell edge forward, such as in the lamellipodium of migrating cells. A single filament generates low forces. However, many growing filaments pushing across an entire protruding cell surface can produce forces three orders of magnitude higher than a single one. In addition, the presence of crosslinkers coupling actin filaments into highly-ordered networks can promote the assembly of stiff structures that support membrane protrusion, such as those building filopodia. Conversely, pulling forces generated by MFs are mediated via their interaction with myosin molecular motors, which slide actin filaments past one another, resulting in the contraction or extension of two bound actin filaments [17,18]. Distinct contractile actin networks assemble at distinct subcellular localizations. Among those, the actomyosin cortex lying under the plasma membrane resists external mechanical stresses, and its local activity drives diverse changes in cell shape, including mitotic cell rounding, cytokinetic furrow ingression, cell body retraction during migration, apical constriction and alterations of epithelial thickness [11,12]. Actomyosin stress fibres can be connected to integrin-mediated focal adhesions (FAs) at their ends (ventral stress fibres) and promote rear constriction during cell migration, or they can be positioned above the nucleus to regulate nuclear shape and to convey forces to it [19], while actomyosin circumferential belts underlying cadherin-mediated Ajs act as direct linkers between adjacent cells to control intercellular surface tension [20].

Although MFs have been for long considered as the predominant engine of force generation, in recent years, multiple reports have demonstrated that MTs also generate physiologically relevant forces controlling cell shape [13,14]. MTs consist of 13 protofilaments composed of αβ tubulin dimers arranged in ∼25 nm wide, polarized hollow tubes. They undergo alternative phases of rapid assembly and disassembly in a process called dynamic instability. This highly dynamic property allows cells to quickly adopt new spatial reorganization, essential to a number of cellular functions, including mitosis. MTs are the stiffest of the three main cytoskeletal polymers [21]. Similar to actin filaments, the addition of new tubulin dimers to MTs in contact with an object, such as the cell cortex, can exert a pushing force, which increases with the number of MTs and the presence of motor and non-motor crosslinkers. Conversely, depolymerization of MTs that remain connected to an object will pull on it. In addition, MT sliding by molecular motor proteins or crosslinkers lacking intrinsic motor activity generates pushing and pulling forces [13].

IFs, similar to MFs and MTs, are also major integrators of cell and tissue mechanics. As opposed to MFs and MTs, which are composed of actin and tubulin, respectively, IFs are formed by one or more members of a large family of highly insoluble proteins encoded by more than 70 genes, whose expression varies between cell types and tissues. IFs can be crosslinked to each other. However, unlike MFs and MTs, they are not polarized and cannot support the directional movement of motor proteins. Instead, it is the fundamental structure of IFs that determines their mechanical properties. Although the flexibility of IFs varies with their composition, single filaments are much more flexible than MFs and MTs. They can be stretched by up to 3 times their original length, whereas MFs and MTs tend to break before being stretched 1.5 times their resting length. Thus, although IF networks tend to be softer than MT and MF networks at low strain, they can withstand much larger deformations. IFs are therefore generally believed to provide mechanical strength and resilience to cells at large deformations [15]. In recent years, it has become clear that while all three cytoskeleton elements can generate forces on their own, they also engage in extensive crosstalk crucial for changes in cell shape [22]. 

## 3. Cell Shape-Dependent Effects on Cell Behaviours

Cell shape has long been recognized to profoundly impact cell behaviour [23]. One behaviour that has been extensively studied is cell proliferation. Untransformed non-lymphoblast-like cells lose their capacity to proliferate when detached from a substrate. The differential proliferative ability of attached versus detached untransformed cells is associated with a striking change in cell shape, characterized by cell rounding in suspension and flattening when anchored to a substrate [24,25]. The degree of ECM-dependent integrin occupation and clustering has been shown to play central roles in the cell response to anchorage [26]. Yet, anchorage per se might not be sufficient to explain these distinct behaviours, as fibroblasts attached to glass beads lose their flat shape and their ability to proliferate when reducing the bead diameter, although they remain anchored [27]. Moreover, stimulating integrin signalling in suspended cells does not trigger proliferation [28,29,30], suggesting that it is the change in cell geometry associated with cell spreading which induces cell division. To test this possibility, micro-patterned cell culturing with substrates containing ECM-coated adhesive islands of defined shapes, sizes and ECM molecular coating densities have been extensively used [31]. These studies demonstrated that cell proliferation in the presence of growth factors depends directly on cell shape, independently of the degree of ECM binding. For example, restricting the spreading of capillary endothelial cells grown on micro-patterned substrates of reduced size while maintaining constant the total cell-matrix contact area restricts cell division [29,32]. The same effect is observed when the actin cytoskeleton or the cytoskeletal tension are disrupted, even though cells remain anchored to the ECM [32]. Similarly, keratinocytes spread on large circular adhesive islands can proliferate. In contrast, DNA synthesis is inhibited when reducing the adhesive area until cells reach an almost spherical shape [33,34]. At the tissue level, cytoskeletal tension has been shown to be directly involved in generating the patterns of proliferation in monolayers of endothelial cells [35]. Furthermore, stretching of epithelial tissues, while preserving cell–cell junctions, leads to an immediate increase in cell area and in the fraction of cycling cells. Conversely, compression induces cell cycle arrest [36,37,38]. Accordingly, during epithelial morphogenesis, local changes in cell shape and cytoskeletal distortion often precede rather than follow changes in proliferation. These architectural changes could allow cells to respond to mitogenic cues locally, thereby further propagating tissue expansion in these regions, as proposed during budding of the lung epithelium [39]. Although these observations cannot rule out that cell proliferation is controlled by local alterations of the ECM or distortion of specific membrane-associated receptors rather than by modifications of the cytoskeleton and lipid bilayer, they suggest that the control of cell proliferation by cell shape holds true in vivo.

It has also become clear that changes in cell shape and mechanical forces instruct cell migration, division and tissue morphogenesis during development [18,40,41]. For instance, stomodeal cell compression during *Drosophila* embryogenesis, likely resulting from the extension of the germ-band, triggers the formation of the anterior gut [42]. Changes in cell shape also have a major impact on cell fate decisions. Thus, growing keratinocytes on smaller circular adhesive islands not only inhibits DNA synthesis but also stimulates their terminal differentiation independently of ECM composition or density [33,34]. Conversely, stretching keratinocytes inhibits their terminal differentiation program [43]. One could deduce that cell stretching, as opposed to rounding, promotes a proliferative undifferentiated state. However, elongated mesenchymal stem cells (MSC) undergo smooth muscle differentiation on large micro-surfaces, while rounded cells on smaller micro-surfaces remain undifferentiated. Moreover, spreading of mouse embryonic stem cells induces their exit from a naive pluripotent state to acquire a primed pluripotent phenotype [44,45]. Cell elongation is also required for myogenesis, as smooth muscle differentiation fails to occur when cells are prevented from elongating in the presence of factors that stimulate their differentiation [46]. Moreover, human MSCs (hMSCs) grown onto large islands of fibronectin have been shown to flatten and differentiate into bone, whereas on small islands, they acquire a round shape and differentiate into fat. These effects depend on a synergy between cell shape and RhoA activity, which both affect intracellular contractility [47]. Furthermore, restricting hMSC spreading or inhibiting cytoskeletal tension prevents their differentiation into bone cells [48]. In addition to the strength of cell spreading, the degree of anisotropy and cell curvature could determine lineage commitment. hMSCs spread on rectangular or star-shaped islands show higher intracellular contractility and preferentially differentiate to bone. In contrast, they undergo adipogenesis on square or concave edge shapes of the same area [49,50]. Taken together, these observations indicate that changes in cell shape can reprogram cell behaviour in a switch-like manner. This comportment is reminiscent of the behaviour of stable attractors of signalling regulatory networks proposed by Stuart Kauffman (see below basins of attraction) [9]. 

## 4. Cell Shape-Dependent Mechanisms Regulating Cell Behaviours

Cell shape changes and cell mechanics have major effects on the transcriptional program [51,52]. For instance, compressive forces or micro-patterned substrates that alter the cellular geometry of mouse fibroblasts induce the nuclear shuttling of histone deacetylase 3, which causes a global increase in chromatin condensation levels and cellular quiescence [53,54]. Moreover, stretching of human epidermal progenitor cells triggers H3K27me3-mediated silencing of nearly 4000 lineage-specific genes [43]. However, the exact molecular mechanisms by which cell shape change affects the transcriptional state of cells to drive distinct cell behaviours are still poorly understood. 

Altering the shape of NIH3T3 fibroblasts using micro-patterned surfaces can directly alter the physical properties of the nucleus via cytoskeletal physical links (e.g., apical stress fibres/perinuclear stress fibres) transmitted from cell surface receptors to the nuclear envelope [55]. As proteins of the inner nuclear membrane and of the nucleoskeleton bind to chromatin, forces transmitted from the cell surface to the nuclear envelope can alter chromatin organization, the accessibility of transcription factors and other chromatin regulatory factors to DNA, and consequently gene expression, as reported in mouse fibroblasts [56,57] (Figure 1a). For instance, nuclear morphological alteration of circle-shaped MC3T3-E1 osteoblasts on micro-patterned islands upregulates the two major calcium cycling proteins, inositol 1,4,5-triphosphate receptor 1 (IP3R1) and sarco/endoplasmic reticulum Ca^2 +^-ATPase 2 (SERCA2), which in turn stimulate cell proliferation [58]. Nuclear flattening as a result of cell shape change can also stretch nuclear pores, reducing their mechanical resistance to the transport of molecules (Figure 1a), as shown for the transcriptional coactivator YAP (Yes-associated protein) in mouse embryonic fibroblasts [59]. Accordingly, diverse reports point to a role of cell shape and mechanical tension in regulating the activity of the two downstream effectors of the Hippo pathway, YAP and TAZ (transcriptional coactivator with PDZ-binding motif) [60]. Primary neonatal human keratinocytes that are forced to acquire a rounded shape on small micro-printed islands exclude YAP from the nucleus and undergo terminal differentiation [61]. Conversely, YAP/TAZ translocate to the nucleus in spread hMSCs or lung microvascular endothelial cells grown on large micro-patterned islands independently of the degree of ECM binding, indicating that cell shape, rather than the strength of cell-ECM interaction, controls YAP/TAZ localization [62]. In addition to promoting YAP nuclear import via alterations of the physical properties of the nucleus, changes in cell shape also regulate the activity of the upstream YAP/TAZ regulators [60]. YAP is mostly cytoplasmic in round fibroblasts grown on small micro-patterned adhesive areas. In contrast, on larger cell adhesive areas, these cells acquire a flat shape and accumulate YAP in the nucleus through a process which depends on the LATS kinase [63].

Alteration of surface tension, bending of the membrane or forces applied on F-actin beneath the plasma membrane can mechanically activate calcium channels (Figure 1b) [64,65]. One such example is the opening of the Piezo1 calcium channel following mechanical stimulations, such as stretching, in diverse mouse and human tissues and cell types [64,66,67]. In turn, calcium currents induce the phosphorylation of the calcium-activated target of MEK1/2, ERK1/2, which triggers cell division in the Madin Darby Canine Kidney (MDCK) cell line [38]. Stretches applied on the plasma membrane can also mechanically unfold or distort proteins leading to exposure or hiding of substrate sites, as demonstrated for the nonreceptor tyrosine kinase c-SRC using molecular dynamics simulations and mouse, primate and human cell lines (Figure 1c) [68,69,70]. Mathematical models and experimental validations using monkey fibroblast cells have also provided evidence that high curvature of membrane micro-domains enhances the activity of membrane-associated receptors following ligand binding [71]. Alteration of lipid raft micro-domains within the cell plasma membrane can also activate signalling pathways in a ligand-independent manner (Figure 1d), as illustrated by the recruitment and activation of Akt, which induces hMSC differentiation [72]. Alterations in cell shape also impinge on the rate of endocytosis. For instance, during the early differentiation of embryonic stem cells, cell spreading decreases tension at the plasma membrane, which stimulates the endocytosis of FGF components and subsequently ERK activation (Figure 1e) [44]. Cell shape-dependent alterations of the ratio between filamentous (F) and globular (G) actin also have major effects on the activity of signalling pathways and epigenetic factors. An increase in the F/G actin ratio in rounded keratinocytes induces the dissociation of the Myocardin-related transcription factor A (MRTF-A) from G-actin, which causes MRTF-A translocation to the nucleus, where it binds to serum response factor (SRF) and activates the expression of target genes supporting terminal differentiation [33]. Altogether, these observations highlight the key role of cell-intrinsic shape determinants in regulating gene expression and cell behaviour.

To explain cell fate switching driven by changes in cell shape, specific signalling pathways have been identified. Accordingly, protein–protein interaction maps indicate that cytoskeleton proteins are closely interconnected with signalling components [73,74]. However, to switch its fate efficiently, a cell must coordinate its response by turning off its current gene program, while turning on a distinct one. This implies that cell shape affects many signalling molecules at once within the whole interconnected signalling network, which converges towards specific cell fate or stable attractors of the network [75,76]. This concept, first introduced by Stuart Kauffman, proposes that in the presence of a stimulus, the internal state of the cell will be attracted into a “basin of attraction” due to the dynamic constraints imposed by the regulatory interaction network, similar to a ball rolling over a landscape with hills and valleys which becomes stabilized in one of the valleys, as illustrated by the epigenetic landscape described by Waddington to explain cell fate decision (Figure 2) [9,77]. Thus, in Waddington’s epigenetic landscape, each basin of attraction could be determined by the constraint imposed by cytoskeleton genes and therefore by cell shape. These shape-dependent constraints could drive the rolling down of cells in specific valleys or their climbing up on the landscape and falling down into other basins of attraction [78]. The cell’s ability to climb up hills and fall down into other valleys could depend on the depth and climbing distance of the initial basin of attraction and therefore the stability of each phenotype. As cells might only be capable of maintaining a set of discrete shapes [8], stable attractors of signalling regulatory networks must, in turn, alter the expression and/or activity of structural components that would stabilize cells into defined shapes and therefore phenotypes.

## 5. Implications to Cancer Initiation and Progression

Because cell shape is a key determinant of cell behaviour, it is therefore not surprising that alterations in cell architecture is also associated with carcinogenesis. Cancer cells display drastic alterations in their shape, cytoskeletal architecture, intracellular tension and force-generating capabilities [79]. Moreover, the expression profile of genes that encode for cytoskeleton regulators differs between normal and cancer cells [80,81]. In addition, image-analysis approaches combined with gene expression and protein–protein interaction data have established correlations between the shape of cancer cells and the activation of signalling pathways [82,83]. Stereotypical cell shapes can also predict cancer cell behaviours, including metastatic potential and chemoresistance, opening the possibility of using morphological parameters to diagnose tumour malignancy and aggressiveness [84,85,86,87,88].

Oncogenes and tumour suppressors are well known to affect the expression of cytoskeleton genes, cell shape and cell mechanics. For instance, expressing a mutated form of the RAS oncogene in the untransformed MCF10A cells induces a striking change in cell shape from flat to cuboidal [89]. These morphological and mechanical alterations are not only byproducts of aberrant cell signalling but actively contribute to driving cancer cell behaviours, including sustaining proliferation, rewiring metabolic pathways, promoting epithelial to mesenchymal transition and propelling metastasis [90,91,92]. Thus, a RAS-dependent increase in cell contractibility promotes tumour progression and aggressiveness [93,94]. In addition, RAS further enhances the rounding and stiffening of mitotic cells to enable cell proliferation under conditions of mechanical confinement, such as those experienced by cells in crowded tumours [95]. We have also shown that conditional activation of an oncogenic form of SRC triggers ERK-dependent proliferation and malignant transformation in MCF10A cells through a transient increase in cell stiffening [96]. In turn, increased contractility of tumour cells and their associated stromal fibroblasts induces tension-dependent ECM remodelling, which feedback on cell architecture [94,97]. Thus, tumorigenic triggers affect cell shape in order to adapt cell geometry to particular cancer cell behaviours.

In addition, the initial shape of untransformed cells might also dictate whether cells are more susceptible to initiate tumorigenesis. Not all cells within a tissue respond equally to tumorigenic triggers. In the *Drosophila* wing imaginal disc, the behaviour of clones of cells depleted of the neoplastic tumour suppressor gene *lethal giant larvae (lgl)* or *scribble (scrib)* depends on the location of the pro-tumour cells. Mutant cells located in the most central blade domain, formed of tall columnar cells, extrude basally and undergo apoptosis. In contrast, in the peripheral hinge region, composed of shorter epithelial cells, mutant clones delaminate apically and show dysplastic growth. These region-specific behaviours correlate with the differential accumulation of apical and basal MTs and depend on the cell-intrinsic architecture, as clones in the blade domain knocked down for *lgl* or *scrib* and *p115 RhoGEF* accumulate basal MTs and extrude apically, similarly to *lgl* or *scrib*-depleted clones localized in the hinge domain [98]. Likewise, in mice, the behaviour of clones expressing an oncogenic form of *KRAS* and mutants for the *Fbw7* tumour suppressor depends on their location within pancreatic ducts. In smaller ducts composed of elongated cells, mutant clones evaginate basally away from the duct lumen and display aggressive behaviour. In contrast, in large ducts formed of cuboidal cells, pro-tumour clones invaginate apically towards the duct lumen and display less aggressive behaviour. These differential responses are not tissue specific, as they can be recapitulated in ducts of the lung. They are also independent of the oncogenic combination, as clones expressing oncogenic *KRAS* and deleted for the *p35* tumour suppressor display identical behaviours. Computational simulations suggest that changes in apical–basal tension in transformed cells and the bending modulus that mimics small and large ducts, dictate the behaviour of mutant clones. Thus, differences in tension imbalance and whole tissue curvature appear to determine tumour morphology and aggressiveness [99]. Even though these observations do not exclude that the interaction of pro-tumour cells with their wild-type neighbours [100] or that fluctuations in the composition and/or organization of the local ECM contribute to the differential behaviour of pro-tumour cells [5], they open the possibility that depending on the cell and/or tissue initial shapes, the same oncogenic trigger can lead to distinct cell fate decisions. This could contribute to explain why tumours are frequently observed at specific locations, for instance at the border between two different types of epithelia, including between the cervix and the uterus, between the oesophagus and the stomach or between the stratified squamous epithelium of anal skin and mucosal epithelium of the large intestine [101,102].

Anomalies in cell shape, cytoskeleton structure and dynamics could also provide a favourable ground for tumour initiation. Accordingly, altering the expression of cytoskeleton genes triggers or predisposes to tumorigenesis. For instance, inactivating actin-capping proteins or overexpressing the actin nucleator *diaphanous* in *Drosophila* epithelia induces tissue overgrowth through activation of the YAP/TAZ oncogene Yorkie [103,104]. The actin, MT and IF crosslinker *Dystonin* also restricts YAP activity and consequently cell growth, anchorage-independent growth, self-renewal and resistance to doxorubicin in the untransformed human mammary epithelial cell line MCF10A [105]. Moreover, conditional knockout of *myosin IIa* (*Myh9*) using an epithelial-specific *Keratin 14*-Cre recombinase induces multiple invasive squamous cell carcinoma in *TGF**β**-Receptor II* conditional knockout mice, suggesting that a decrease in cell mechanics enhances tumour susceptibility [106]. Change in cell shape could also be one of the biological alterations associated with ageing that could contribute to explaining the increase in cancer incidence over the years [107]. Cells display drastic alterations of their shape during ageing, as observed in the epidermis [108]. Cellular ageing is also associated with significant cytoskeleton alterations [109,110]. Actin levels per se go significantly down in old rats and *Drosophila* when compared with younger individuals [111,112]. Despite a reduction in actin levels, many different cell types, including skin epithelial cells, show increases in actin filament polymerization and cell stiffening with ageing, leading to reduced cell elasticity [113,114]. In *Caenorhabditis elegans*, actin cytoskeletal disorganization resulting from ageing can be prevented by overexpressing the heat shock factor (HSF-1). Conversely, inhibiting HSF-1 function causes the actin cytoskeleton to age prematurely [115]. Ageing is also associated with alterations of the MTs. They become disrupted and disorganized in *Drosophila* cells and are fewer and/or shorter in old rats [116,117]. Alterations in the expression of IFs during ageing are also associated with changes in cell elasticity during ageing, as observed in skin fibroblasts [118]. However, a causal link between change in cell shape and ageing or between age-associated change in cell shape and cancer incidence is still missing.

Thus, when considering the Waddington’s epigenetic landscape, in the presence of an oncogenic trigger, the ability of cells to climb up hills and fall into distinct basins of attraction could depend on the constraints imposed by cell shape. Depending on the stability of each attractor (depth and climbing distance of the basin), mutations in genes affecting the cytoskeleton’s structure and dynamics could also reprogram the whole signalling network, allowing cells to reach other basins of attraction (Figure 2). However, fully transformed cells display complete loss of shape-dependent growth (anchorage-independent growth) and shape-responsive metabolic control [23,119,120]. Thus, to progress towards a fully transformed phenotype, cells might need to escape the restrictions imposed by cell shape. This could allow cancer cells to move into the landscape more easily and to reach abnormal basins of attraction that are distinct from those occupied by untransformed cells (Figure 2), as conceptualized by Stuart Kauffman [121,122]. The breaking down of these shape-dependent constraints could support cancer cell plasticity, thus allowing cells to survive and adapt to hostile microenvironments.

## 6. Conclusions

Taken all together, these observations provide evidence that cell shape, dictated by the organization and mechanical properties of the cytoskeleton, has a central role in controlling the behaviour of normal and pro-tumour cells. To coordinate cell fate switching, global and local changes in cell shape affect many signalling molecules simultaneously within the whole interconnected signalling network. Because the cytoskeleton constitutes a coherent, unified system running all over the cytoplasm and linking a variety of cellular compartments, including AJs and FAs or the plasma membrane and the nuclear envelop, it fits into a suitable position to coordinate signalling events and therefore the cell response through changes in gene transcription. The cell shape-dependent control on signalling networks can help to explain why alterations in the expression or activity of cytoskeletal components can drive cancer-associated phenotypes or may provide a favourable ground for tumorigenesis in the presence of an oncogenic trigger, such as in elderly cells. It can also explain why tumours arise in specific locations. Thus, a robust control of the cell shape machinery may represent a critical safeguard against tumorigenesis that needs to be neutralized by cancer cells in order to survive assaults from the microenvironment. This raises the question of how pro-tumour cells could override these shape-dependent constraints. Further work is required to elucidate the rules underlying the control of regulatory networks by the cytoskeleton in normal cells and how this control is affected in cancer cells.

## Figures and Tables

**Figure 1 ijms-23-08622-f001:**
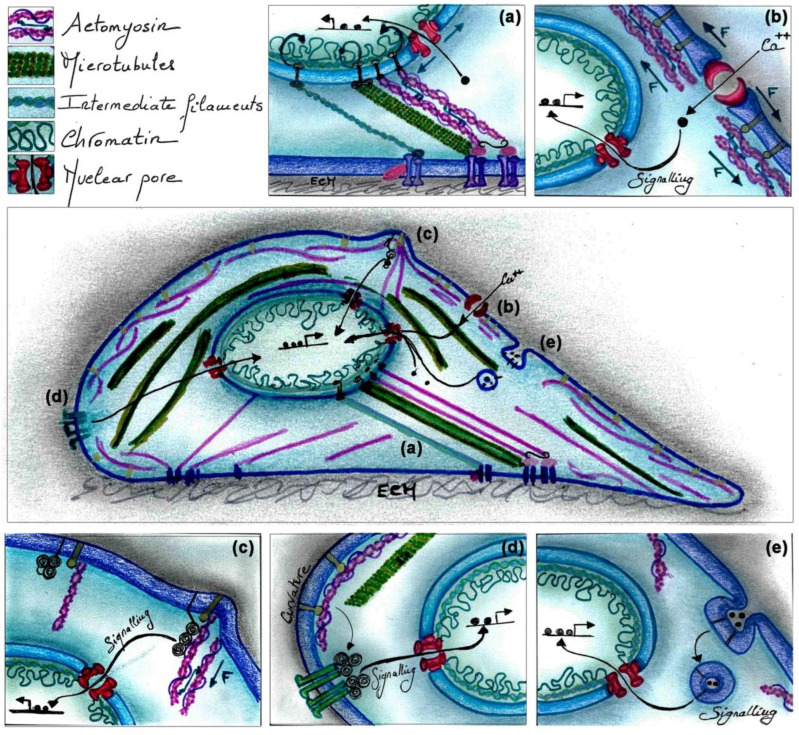
**Examples of mechanisms by which cell shape regulates gene transcription.** (**a**) Forces transmitted from the cell surface to the nuclear envelope through MFs, MTs or IFs can alter chromatin organization, the accessibility of transcription and other chromatin regulatory factors, as well as stretch nuclear pores to facilitate the transport of molecules. (**b**) Alteration of surface tension, bending of the membrane or forces applied on F-actin beneath the plasma membrane can mechanically activate calcium channels. (**c**) Stretching of the membrane can also induce tension on actomyosin, which can unfold proteins, exposing their substrate sites and induce signalling to the nucleus. (**d**) Cell shape-dependent alterations of the membrane and of the actin cytoskeleton can alter lipid raft micro-domains within the plasma membrane and activate signalling pathways. (**e**) Shape-dependent alteration of cell surface tension can activate signalling pathways by stimulating the rate of endocytosis.

**Figure 2 ijms-23-08622-f002:**
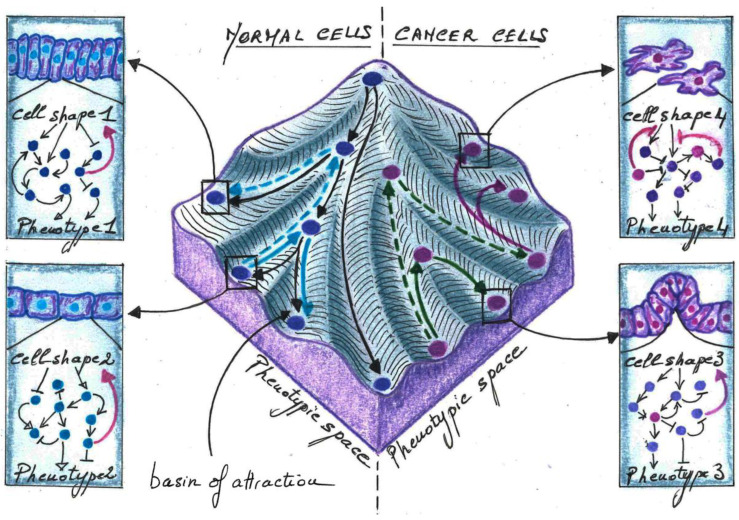
**Schematic inspired by Waddington’s epigenetic landscape depicting how cell shape could act as an attractor to control the fate of normal and cancer cells.** (**Middle landscape**) Each ball moving into the valleys represents a cell defined by its specific shape and network state represented in the boxes on the left and right of the landscape. The blue balls on the left are normal cells. The pink balls on the right are cancer cells. (**Left**) The shape of normal cells determines which valley will be taken by the cell (black arrows) to reach specific basins of attraction. Cell shape 1 and 2 control specific regulatory interaction networks, which trigger phenotype 1 and 2, respectively. Each regulatory network also stabilizes cell shapes 1 and 2, respectively (pink arrows). Changes in cell shape could trigger cell reprogramming by which a cell climbs up on the landscape and falls down into another basin of attraction (plain blue arrows). These cell shape-dependent phenotypic reversions could rely on the climbing distance and depth of the initial basin of attraction. If it is modest, reversion will be easily attainable (plain blue arrows). However, if it is important, reversion would be less likely (dashed blue arrows). (**Right**) Cells suffering anomalies in their shape (cell shape 3 and 4) and/or acquiring mutations in oncogenic signalling components (pink spots) could climb up the hill and fall down into other basins of attraction (plain green arrow), for example by reacquiring proliferative abilities. To attain basins of attraction that require higher distances to climb up, cells may need to acquire additional cellular alterations (dashed green arrows). During cancer progression, alterations of the regulatory network could inhibit the cell shape-dependent control (pink inhibitions in the upper right box). By escaping the restrictions imposed by cell shape, cells could reach new attractors not attained by normal cells (pink arrows).

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
