# Peer review of "Cell Architecture-Dependent Constraints: Critical Safeguards to Carcinogenesis"

_ijms, 2022, doi:10.3390/ijms23158622_

Round 1

Reviewer 1 Report

In their manuscript "Cell architecture-dependent constrains: Critical safeguards to carcinogenesis", Khalil, Eon and Janody present (as I understood it) the idea that cellular architecture influences gene regulatory networks just as much as it is influenced by these. And therefore, when considering cells that move from one phenotype to another (particularly in the context of tumour initiation and progression) these transitions are as constrained by the mechanical and morphological properties of cells as by the topology of their gene regulatory networks. The manuscript is quite easy to read, the hand-drawn illustrations are very nice and refreshing (if not very colourblind-friendly), and overall I find it worth publishing with minor adjustments.   I guess it could be said that the ideas presented in the manuscript are not radically new. The observation that mechanical context and cellular shape constrain proliferation, differentiation, and gene expression have been known since at least the 90s -- although some understanding of the mechanisms behind this is more recent (some random reviews: PMIDs 32671813, 33712293). Also, the idea of regulatory network constrains being "jumped up" by tumour cells as switches between attractors was proposed by Sui Huang and Stuart Kauffman a good decade and a half ago ("cancer attractors", PMID: 19595782, 23792873). In my mind, this manuscript puts these two ideas together, but I am not aware that this has been done explicitly before. There is value in this and, therefore, I repeat that the review merits publication (with some acknowledgment of the above). (This is assuming I have understood the ideas correctly; if not, then I guess the manuscript is presenting a something more sophisticated, and I guess this will make it more valuable -- but the writing less clear).   Whether I have understood it correctly or not, I think that, before the manuscript is published, a bit of work on its clarity/additional detail would help: - The first thing to mention would be the actual tenet of the manuscript: there are points when it seems to be focusing on cell shape (rather than cell mechanics), but there are others where tension and contractility seem to be more central. Being more explicit and direct about what the take home message is would be very helpful. - Another area that is a bit obscure is the review of the mechanistic connection between shape and mechanics (section 4): there is a long list of observations, but very little integration, organisation, or extraction of common themes (e.g. tension-dependent modification of protein function/structure). - Section 5, where the connection with cancer is made, is a bit confusing, as it starts stating a series of correlations between cell shape and cancer, without leaving it clear whether the view of the authors is that shape determines cancer behaviour, or the other way around. I guess it is the former view, but all those observations require the prescriptive warning that it is difficult to determine what causes what. Another matter that warrants some discussion (or at least prescriptive warnings) is the cell autonomy of some of the observations. In particular, p/c line 339 does not get to discuss whether the differential behaviour of the clones is cell-autonomous or not (in the case of Drosophila, it is, but this is not stated explicitly; in the case of the mouse, this cannot be established from the reported observations). This is important, as the work around cell competition-induced transformed-cell extrusion from epithelia (the so-called "EDAC") clearly states that those behaviours are dependent on cell-cell interactions, and non-transformed neighbour cell behaviour is as important as the transformed ones. - Also in Section 5, there are some lists of observations where the interpretation, in the eye of the authors, is not clear. For instance, in lines 375-6: "the function of myosin IIa induces invasive squamous cell carcinomas on tumour-susceptible backgrounds in mice"; do they interpret here that is tension, or cell shape, that induces invasiveness? And is this a cell autonomous property? (I am assuming human cancer cell lines are treated in vitro, then injected into immunocompromised mice, but this could be a systemic treatment in the mouse, or a conditional genetic intervention in some mouse tissue...). Like this, there are other examples where it is not immediately clear what the authors say is going on. - In several places, it is not clear in which research system were the observations made. For instance, lines 233-246, several experiments are reported, but it is not clear whether the observations are all coherently made in one system, or have been put together from several different organisms/cell types - this is important, as it influences the interpretation.   A few other, smaller things: - The level of previous knowledge expected from the reader is not homogeneous across the text. For instance, there is a quick review of very basic aspects of the properties of microfilaments, microtubules and intermediate filaments (e.g. "These highly dynamic structures [MT] display growth and shrinkage in a process called dynamic instability"), but then the reader is expected to know how integrins work, what are the usual components of the extracellular matrix, or how to compare "stretching 3 times its length" with a "strain of 50%". Some thought of how much the target audience knows would help here. - Line 377+: "Change in cell shape could also be one of the biological alterations associated with aging that could contribute to explain the increase in cancer incidence over the years [102]. Indeed, skin epidermal cells display drastic alterations of their shape during aging [103]. Cellular aging is also associated with significant cytoskeleton alterations [104,105]." This warrants the usual cautionary note of causation vs correlation. - Line 111: perhaps add a line quickly explaining what dynamic instability is? Just to be on par with the introduction to microfilaments, in level of detail. - Line 119: 'IFs are also major integrators...' it sounds as if you had been talking about the function of IFs before. I suggest to rephrase: "IFs, like MFs and MTs, are also..." - Lines 126-8: "They can be stretched by up to 3 times their original length, whereas MFs and MTs tend to break at strains of less than 50%". I would rephrase by saying: "... whereas MFs and MTs tend to break before being stretched 1.5 times (their resting length)" or something to that effect. - Line 135: "One behaviour that has been extensively studied, is cell proliferation." Comma is not needed. - Line 136: "Untransformed cells lose their capacity to proliferate when detached from a substrate" - maybe specify that this is for the non-lymphoblast-like cells? - Line 137: "differential behaviour" is comparing attached vs detached cells, or transformed vs non-transformed? - Line 130: Integrins have not been introduced as much as IF/MT/MFs and they are mentioned quite often. - Line 146: "micropatterns culture substrates" - should be "micro-patterned..."? - Line 217: "as shown for the transcriptional coactivators YAP (Yes-associated protein) [58]." Coactivator, singular? - Line 246: "membrane-to-cortex detachments" - not clear what this means: MF cortex detached from plasma membrane? That's not what is shown in Fig 1-5. - Line 250: - "the dissociation of the Myocardin-related transcription factor A (MRTF-A) from G-actin, which translocates to the nucleus" - here, "which" seems to refer to G-actin, not MRTFA.

Author Response

Reviewer 1: In their manuscript "Cell architecture-dependent constrains: Critical safeguards to carcinogenesis", Khalil, Eon and Janody present (as I understood it) the idea that cellular architecture influences gene regulatory networks just as much as it is influenced by these. And therefore, when considering cells that move from one phenotype to another (particularly in the context of tumour initiation and progression) these transitions are as constrained by the mechanical and morphological properties of cells as by the topology of their gene regulatory networks. The manuscript is quite easy to read, the hand-drawn illustrations are very nice and refreshing (if not very colourblind-friendly), and overall I find it worth publishing with minor adjustments.   I guess it could be said that the ideas presented in the manuscript are not radically new. The observation that mechanical context and cellular shape constrain proliferation, differentiation, and gene expression have been known since at least the 90s -- although some understanding of the mechanisms behind this is more recent (some random reviews: PMIDs 32671813, 33712293). Also, the idea of regulatory network constrains being "jumped up" by tumour cells as switches between attractors was proposed by Sui Huang and Stuart Kauffman a good decade and a half ago ("cancer attractors", PMID: 19595782, 23792873). In my mind, this manuscript puts these two ideas together, but I am not aware that this has been done explicitly before. There is value in this and, therefore, I repeat that the review merits publication (with some acknowledgment of the above). (This is assuming I have understood the ideas correctly; if not, then I guess the manuscript is presenting a something more sophisticated, and I guess this will make it more valuable -- but the writing less clear).   Whether I have understood it correctly or not, I think that, before the manuscript is published, a bit of work on its clarity/additional detail would help:

Response: We thank the reviewer for these positive comments and for the time taken to help us improving the content and grammar of our review. As mentioned by the reviewer, we aim to discuss the idea that cell shape is a main regulator or attractor of cell behaviour in normal and cancer cells through the control of signalling networks. To join these ideas together, we review data reporting how cell shape constrains cell behaviour in untransformed cells, discuss the idea that cell shape could act as a critical safeguard to tumour initiation and progression and how in turn, cancer cells could escape the restrictions imposed by cell shape to undergo malignant transformation, using the concept of the epigenetic landscape proposed by Stuart Kauffman. We also acknowledge the work of Donald E. Ingber, who proposed that cell shape controls the regulatory network in the context of normal development [1]. We have acknowledged the publications mentioned by the reviewer.

Reviewer 1: - The first thing to mention would be the actual tenet of the manuscript: there are points when it seems to be focusing on cell shape (rather than cell mechanics), but there are others where tension and contractility seem to be more central. Being more explicit and direct about what the take home message is would be very helpful. – Another area that is a bit obscure is the review of the mechanistic connection between shape and mechanics (section 4): there is a long list of observations, but very little integration, organisation, or extraction of common themes (e.g. tension-dependent modification of protein function/structure).

Response: In the introduction, we define cell shape as follow: “In this review, we discuss the role of cell shape, as a direct readout of intrinsic forces exerted by the cytoskeleton on the cell membrane, in normal, pro-tumour and cancer cells.” We made clear in the text when experimental evidences were obtained by altering cell shape using micro-patterned substrates, the cytoskeleton or cell mechanics.

Reviewer 1: - Section 5, where the connection with cancer is made, is a bit confusing, as it starts stating a series of correlations between cell shape and cancer, without leaving it clear whether the view of the authors is that shape determines cancer behaviour, or the other way around. I guess it is the former view, but all those observations require the prescriptive warning that it is difficult to determine what causes what.

Response: We start section 5 by highlighting correlations between changes in cell shape and carcinogenesis. We then describe in the second paragraph evidences reporting how tumorigenic triggers alter cell shape to promote cancer cell behaviours. In the third and fourth paragraph, we aim to argue that the initial shape of untransformed cells might also dictate the cell capacity to initiate tumorigenesis. We first report evidences suggesting that the response of cells to an oncogenic trigger might depend on the cell and/or tissue initial shapes (third paragraph). We then provide evidences that alterations in cell shape, resulting from aging or from mutations in cytoskeletal regulators could provide a favourable ground for tumour initiation (fourth paragraph). We have tried to clarify these ideas by concluding at the end of the second paragraph as follow: “Thus, tumorigenic triggers affect cell shape in order to adapt cell geometry to particular cancer cell behaviours.” and by starting the third paragraph as follow: “In addition, the initial shape of untransformed cells might also dictate whether cells are more susceptible to initiate tumorigenesis.”

Reviewer 1: Another matter that warrants some discussion (or at least prescriptive warnings) is the cell autonomy of some of the observations. In particular, p/c line 339 does not get to discuss whether the differential behaviour of the clones is cell-autonomous or not (in the case of Drosophila, it is, but this is not stated explicitly; in the case of the mouse, this cannot be established from the reported observations). This is important, as the work around cell competition-induced transformed-cell extrusion from epithelia (the so-called "EDAC") clearly states that those behaviours are dependent on cell-cell interactions, and non-transformed neighbour cell behaviour is as important as the transformed ones.

Response: In the work from Messal et al., the authors analyse the behaviour of mutant clones marked with a membrane-tethered GFP lineage label in small and large pancreatic ducts and show that their differential behaviours correlate with the size of the duct. To identify the parameters that determine these differential behaviours, they built a computational model of the pancreatic duct using a 3D vertex model simulation, which integrates duct cell geometry, apical, basal and lateral surface tensions, and cell volume conservation. Model simulations indicate that the magnitude of the changes in apical–basal tension in the transformed region and the bending modulus of the tissue determine the behaviour of mutant clones in small and large ducts [2]. Thus, even though these observations do not exclude a role of cell interaction with neighbouring wild type cells, they suggest that differences in tension imbalance and whole tissue curvature in small versus large ducts determine tumour morphology and aggressiveness. We have tried to clarify the text and added and conclude as follow: “Even though these observations do not exclude that the interaction of pro-tumour cells with their wild type neighbours [3] or fluctuations in the composition and/or organization of the local ECM, contribute to the differential behaviour of pro-tumour cells [4], they open the possibility that depending on the cell and/or tissue initial shapes, the same oncogenic trigger can lead to distinct cell fate decisions.

Reviewer 1: - Also in Section 5, there are some lists of observations where the interpretation, in the eye of the authors, is not clear. For instance, in lines 375-6: "the function of myosin IIa induces invasive squamous cell carcinomas on tumour-susceptible backgrounds in mice"; do they interpret here that is tension, or cell shape, that induces invasiveness? And is this a cell autonomous property? (I am assuming human cancer cell lines are treated in vitro, then injected into immunocompromised mice, but this could be a systemic treatment in the mouse, or a conditional genetic intervention in some mouse tissue...).

Response: We have clarified the way these data were gathered and our interpretation as follow: “Moreover, conditional  knockout of myosin IIa (Myh9) using an epithelial-specific Keratin 14-Cre recombinase, induces multiple invasive squamous cell carcinomas in TGF-Receptor II conditional knockout mice, suggesting that a decrease in cell mechanics enhances tumour susceptibility [5].”

Reviewer 1: Like this, there are other examples where it is not immediately clear what the authors say is going on. - In several places, it is not clear in which research system were the observations made. For instance, lines 233-246, several experiments are reported, but it is not clear whether the observations are all coherently made in one system, or have been put together from several different organisms/cell types - this is important, as it influences the interpretation.  

Response: For each experimental evidence presented in this review, we have included the experimental models used when missing.

Reviewer 1: A few other, smaller things: - The level of previous knowledge expected from the reader is not homogeneous across the text. For instance, there is a quick review of very basic aspects of the properties of microfilaments, microtubules and intermediate filaments (e.g. "These highly dynamic structures [MT] display growth and shrinkage in a process called dynamic instability"), but then the reader is expected to know how integrins work, what are the usual components of the extracellular matrix, or how to compare "stretching 3 times its length" with a "strain of 50%". Some thought of how much the target audience knows would help here.

Response: We have briefly introduced the ECM and integrins in the text. Yet, because our review focuses on cell intrinsic forces, we do not think that adding extensive descriptions on ECM and integrins would help to understand the message we would like to convey.

Reviewer 1: - Line 377+: "Change in cell shape could also be one of the biological alterations associated with aging that could contribute to explain the increase in cancer incidence over the years [102]. Indeed, skin epidermal cells display drastic alterations of their shape during aging [103]. Cellular aging is also associated with significant cytoskeleton alterations [104,105]." This warrants the usual cautionary note of causation vs correlation.

Response: As pointed by the reviewer, there are currently no demonstration of a causal link between aging and cell shape, nor between age-associated change in cell shape and cancer. We have therefore added a note of caution at the end of this paragraph as follow: “Yet, a causal link between change in cell shape and aging or between age-associated change in cell shape and cancer incidence is still missing.”

Reviewer 1: - Line 111: perhaps add a line quickly explaining what dynamic instability is? Just to be on par with the introduction to microfilaments, in level of detail.

Response: We have altered this part as follow: “They undergo alternative phases of rapid assembly and disassembly in a process called dynamic instability. This highly dynamic property allows cells to quickly adopt new spatial reorganization, essential to a number of cellular functions, including mitosis.”

Reviewer 1: - Line 119: 'IFs are also major integrators...' it sounds as if you had been talking about the function of IFs before. I suggest to rephrase: "IFs, like MFs and MTs, are also..."

Response: We have replaced “IFs are also major integrators….” by “IFs, like MFs and MTs, are also...”

Reviewer 1: - Lines 126-8: "They can be stretched by up to 3 times their original length, whereas MFs and MTs tend to break at strains of less than 50%". I would rephrase by saying: "... whereas MFs and MTs tend to break before being stretched 1.5 times (their resting length)" or something to that effect.

Response: We have replaced “…..whereas MFs and MTs tend to break at strains of less than 50%” by “…..whereas MFs and MTs tend to break before being stretched 1.5 times their resting length.

Reviewer 1: - Line 135: "One behaviour that has been extensively studied, is cell proliferation." Comma is not needed.

Response: We remove the comma in the sentence "One behaviour that has been extensively studied, is cell proliferation."

Reviewer 1: - Line 136: "Untransformed cells lose their capacity to proliferate when detached from a substrate" - maybe specify that this is for the non-lymphoblast-like cells?

Response: We have replaced “Untransformed cells lose their capacity to proliferate when detached from a substrate.” by “Untransformed non-lymphoblast-like cells lose their capacity to proliferate when detached from a substrate.”

Reviewer 1: - Line 137: "differential behaviour" is comparing attached vs detached cells, or transformed vs non-transformed?

Response: The “differential behaviour….” refers to attached vs detached untransformed cells, as transformed cells acquire anchorage-independent growth abilities. We have rephrased the sentence as follow: “The differential proliferative ability of attached versus detached untransformed cells is associated with….”

Reviewer 1: - Line 130: Integrins have not been introduced as much as IF/MT/MFs and they are mentioned quite often.

Response: We briefly introduced integrins in the text. Yet, because our review focuses on cell intrinsic forces, we do not think that adding an extensive description on how integrins sense extrinsic forces would help to understand the message we would like to convey.

Reviewer 1: - Line 146: "micropatterns culture substrates" - should be "micro-patterned..."?

Response: We have replaced “micropatterns” by "micro-patterned” in line 146, as well as other places in the text.

Reviewer 1: - Line 217: "as shown for the transcriptional coactivators YAP (Yes-associated protein) [58]." Coactivator, singular?

Response: We removed the “s” to coactivators on line 217.

Reviewer 1: - Line 246: "membrane-to-cortex detachments" - not clear what this means: MF cortex detached from plasma membrane? That's not what is shown in Fig 1-5.

Response: Although the differentiation of naïve embryonic stem cells involves the detachment of the actin cortex to the plasma membrane [6], the report from De Belly et al., does not show that membrane-to-actin cortex detachments affects the rate of endocytosis. We have therefore altered this part as follow: “Alterations in cell shape also impinge on the rate of endocytosis. For instance, during the early differentiation of embryonic stem cells, cell spreading decreases tension at the plasma membrane, which stimulates the endocytosis of FGF components and subsequently ERK activation.”

Reviewer 1: - Line 250: - "the dissociation of the Myocardin-related transcription factor A (MRTF-A) from G-actin, which translocates to the nucleus" - here, "which" seems to refer to G-actin, not MRTFA.

Response: We have rephrase this sentence as follow: “…..the dissociation of the Myocardin-related transcription factor A (MRTF-A) from G-actin, which causes MRTF-A translocation to the nucleus,….”.

References

  1. Ingber, D.E. Mechanical control of tissue morphogenesis during embryological development. Int. J. Dev. Biol. 2006, 50, 255–266, doi:10.1387/ijdb.052044di.
  2. Messal, H.A.; Alt, S.; Ferreira, R.M.M.; Gribben, C.; Wang, V.M.Y.; Cotoi, C.G.; Salbreux, G.; Behrens, A. Tissue curvature and apicobasal mechanical tension imbalance instruct cancer morphogenesis. Nature 2019, 566, 126–130, doi:10.1038/s41586-019-0891-2.
  3. Tanimura, N.; Fujita, Y. Epithelial defense against cancer (EDAC). Semin. Cancer Biol. 2020, 63, 44–48.
  4. Hayward, M.K.; Muncie, J.M.; Weaver, V.M. Tissue mechanics in stem cell fate, development, and cancer. Dev. Cell 2021, 56, 1833–1847.
  5. Schramek, D.; Sendoel, A.; Segal, J.P.; Beronja, S.; Heller, E.; Oristian, D.; Reva, B.; Fuchs, E. Direct in vivo RNAi screen unveils myosin IIa as a tumor suppressor of squamous cell carcinomas. Science (80-. ). 2014, 343, 309–313, doi:10.1126/science.1248627.
  6. Bergert, M.; Lembo, S.; Sharma, S.; Russo, L.; Milovanović, D.; Gretarsson, K.H.; Börmel, M.; Neveu, P.A.; Hackett, J.A.; Petsalaki, E.; et al. Cell Surface Mechanics Gate Embryonic Stem Cell Differentiation. Cell Stem Cell 2021, 28, 209-216.e4, doi:10.1016/j.stem.2020.10.017.

Reviewer 2 Report

The review  presented by Khalil et al (Cell architecture-dependent constrains: Critical safeguards to carcinogenesis) collected and summarized the recent reports which show that cell shape has a central role in controlling the behaviour of normal and pro-tumour cells.

the review is very interesting and i have only minor comments:

In the review some grammars mistakes and typos need to be corrected.

Lines 313-314 need to be re-written as it confirms that any change in the cell shape is only related to the carcinogenesis and not to other processes.

Author Response

Reviewer 2: The review  presented by Khalil et al (Cell architecture-dependent constrains: Critical safeguards to carcinogenesis) collected and summarized the recent reports which show that cell shape has a central role in controlling the behaviour of normal and pro-tumour cells.

the review is very interesting and i have only minor comments:

In the review some grammars mistakes and typos need to be corrected.

Response: We went through the text and corrected the grammars mistakes and typos.

Reviewer 2: Lines 313-314 need to be re-written as it confirms that any change in the cell shape is only related to the carcinogenesis and not to other processes.

Response: We have re-written the first sentence of section 5 as follow: “Because cell shape is a key determinant of cell behaviour, it is therefore not surprising that alteration in cell architecture is also associated with carcinogenesis.”

Reviewer 3 Report

This is an excellent review that highlights the importance of cell shape and morphology. The article nicely describes key aspects of intracellular components that are required to drive changes in cell morphology.  It also emphasizes the differences between normal and cancer cells and provides the basis for how some of the restrictions of cell shape control could be lost in cancer cells to migrate and metastasize. 

I suggest the authors also look into some of the anti-cancer therapy approaches that focus on blocking cancer cell migration/metastasis by blocking changes in the cell shape eventually blocking migration. Also, a list of oncogenes (in tabular form) that affects key cellular components that in turn affect cell motility and shape would be ideal.

Author Response

Reviewer 3: This is an excellent review that highlights the importance of cell shape and morphology. The article nicely describes key aspects of intracellular components that are required to drive changes in cell morphology.  It also emphasizes the differences between normal and cancer cells and provides the basis for how some of the restrictions of cell shape control could be lost in cancer cells to migrate and metastasize. 

I suggest the authors also look into some of the anti-cancer therapy approaches that focus on blocking cancer cell migration/metastasis by blocking changes in the cell shape eventually blocking migration. Also, a list of oncogenes (in tabular form) that affects key cellular components that in turn affect cell motility and shape would be ideal.

Response: We thanks the reviewer for these positive comments. We agree with the reviewer that we poorly access cell shape and migration/metastasis in this review. Even though we discuss briefly the role of cell shape downstream of oncogenes in promoting cancer cell behaviours, including migration/metastasis, our goal in this review was to emphasize that cell shape might predispose cells to initiate tumorigenesis, an idea that, to our knowledge, has not called much attention yet.